# Could the mitotic count improve personalized prognosis in melanoma patients?

Alessandra Buja[1], Massimo Rugge[2,3], Claudia Cozzolino[4]*, Francesca Dossi[1], Manuel Zorzi[2], Antonella Vecchiato[4], Giuseppe de Luca[1,5], Paolo Del Fiore[4], Saveria Tropea[4], Luigi dall'Olmo[4,6], Carlo Riccardo Rossi[6], Giovanna Boccuzzo[7], Simone Mocellin[4,6]

**1** Department of Cardiac, Thoracic, Vascular Sciences and Public Health, University of Padova, Padova, Italy, **2** Veneto Tumor Registry, Azienda Zero, Padova, Italy, **3** Department of Medicine—DIMED, Pathology and Cytopathology Unit, University of Padova, Padova, Italy, **4** Soft-Tissue, Peritoneum and Melanoma Surgical Oncology Unit, Veneto Institute of Oncology IOV- IRCCS, Padova, Italy, **5** Directorate General, Veneto Institute of Oncology IOV- IRCCS, Padova, Italy, **6** Department of Surgery, Oncology and Gastroenterology—DISCOG, University of Padova, Padova, Italy, **7** Department of Statistics, University of Padova, Padova, Italy

\* claudia.cozzolino@iov.veneto.it

**Data Availability Statement:** The data supporting this study's findings are held by the Veneto Epidemiological Registry and were used under license for this work. The anonymized minimal data set necessary to replicate our findings has been

## Abstract

A number of studies have indicated that the mitotic rate may be a predictive factor for poor prognosis in melanoma patients. The aim of this study was to investigate whether the mitotic rate is associated with other prognostic clinical and anatomopathological characteristics. After adjusting for other anatomopathological characteristics, we then verified the prognostic value of the number of mitoses, determining in which population subgroup this variable may have greater prognostic significance on 3-year mortality. The Veneto Cancer Registry (Registro Tumori del Veneto—RTV), a high-resolution population-based dataset covering the regional population of approximately 4.9 million residents, served as the clinical data source for the analysis. Inclusion criteria included all incident cases of invasive cutaneous malignant melanoma recorded in the RTV in 2015 (1,050 cases) and 2017 (1,205 cases) for which the number of mitoses was available. Mitotic classes were represented by Kaplan–Meier curves for short-term overall survival. Cox regression calculated hazard ratios in multivariable models to evaluate the independent prognostic role of different mitotic rate cut-offs. The results indicate that the mitotic rate is associated with other survival prognostic factors: the variables comprising the TNM stage (e.g., tumor thickness, ulceration, lymph node status and presence of metastasis) and the characteristics that are not included in the TNM stage (e.g., age, site of tumor, type of morphology, growth pattern and TIL). Moreover, this study demonstrated that, even after adjusting for these prognostic factors, mitoses per mm$^2$ are associated with higher mortality, particularly in T2 patients. In conclusion, these findings revealed the need to include the mitotic rate in the histological diagnosis because it correlates with the prognosis as an independent factor. The mitotic rate can be used to develop a personalized medicine approach in the treatment and follow-up monitoring of melanoma patients.

made publicly available at the following link: https://doi.org/10.6084/m9.figshare.24637938.v1.

**Funding:** This research has received "Current Research 2023" funds from the Italian Ministry of Health to cover publication costs.

**Competing interests:** The authors have no conflicts of interest to declare.

## Introduction

The incidence of cutaneous malignant melanoma (CMM) in white people has steadily increased in recent decades. In Europe, a recent study on melanoma incidence trends revealed a statistically significant increase in incidence for both invasive (+ 4.0% men, + 3.0% women) and in situ (+ 7.7% men, + 6.2% women) cases [1]. In Italy, melanoma is the second most common cancer among males under 50, and the third most prevalent among females under 50 [2,3].

Formal staging of cancer is necessary for prognostic information, developing treatment strategies, and directing and analyzing clinical trials. The staging of cutaneous melanoma continues to evolve through the identification and rigorous analysis of potential prognostic factors. The first multivariate analyses of prognostic factors for melanoma were published over three decades ago [4,5]. Over the past two decades, a greater understanding of cutaneous melanoma has led to the conclusion that many prognostic factors are interrelated [6,7]. Tumor thickness, ulceration, and growth phase serve as prognostic factors for a specific subgroup of melanoma patients (e.g., stage I melanoma) [8]. Since then, several well-designed studies have improved our understanding of the relevant prognostic indicators for this disease, while other factors, such as mitotic rates, require further refinement of prognostic data. Numerous studies have suggested that the mitotic rate (measured as the number of mitoses per square millimeter), which is a quantifiable marker of tumor cell proliferation, could be a predictive factor for poor prognosis in melanoma patients, since a higher mitotic index indicates cells with greater doubling times that are more likely to grow and invade adjacent lymphatic and blood vessels, and thus may be a predictor of poor prognosis [9–11]. Hale recommended including the conventional mitotic rate in diagnostic reports [12], as it correlates strongly with clinical outcomes such as progression-free survival [13,14]. However, further research of melanoma prognostic factors may be a further step towards the emergence of "personalized medicine" (also known as "precision medicine" or "individualized medicine"), which includes diagnostic, preventive, and therapeutic measures that are optimally tailored to an individual. Precision medicine refers to the tailoring of medical treatment to each patient's specific characteristics, involving the ability to classify individuals into subpopulations that are highly responsive to a specific treatment, thereby reducing treatment costs and side effects. Consequently, personalized medicine improves patient outcomes and reduces the need for unnecessary and expensive therapies. This reduces healthcare spending, to the obvious benefit of healthcare professionals and patients.

The aim of this study was to verify, after adjusting for other prognostic clinical and anatomopathological characteristics, the prognostic value of the number of mitoses, determining the population subgroups in which this variable may have prognostic significance in terms of short-term mortality in order to enable a more personalized and precise approach to the management of melanoma patient treatment and follow-up.

## Methods

### Context

The Italian National Health System is a public service founded on the core principles of universality, free access, freedom of choice, pluralism and equity. In terms of organization, it is managed regionally and primarily funded by general taxation [15].

In 2015, the Veneto Oncology Network (Rete Oncologica Veneta—ROV) published a comprehensive document based on the most recent national and international literature detailing the clinical protocols for the clinical management of CMM patients [16–19].

## Materials

The Veneto Cancer Registry (Registro Tumori del Veneto—RTV), a high-resolution population-based dataset covering the regional population of approximately 4.9 million residents, and the regional health service records were used as clinical data sources for the analysis. Cancer registration procedures were based on information acquired from various sources (e.g., pathology reports, death certificates, and the health service's administrative records).

Inclusion criteria were all incident cases of invasive CMM recorded in the RTV in 2015 (1,050 cases) and 2017 (1,205 cases) for which the number of mitoses was available. The following variables were considered in this study: sociodemographics (age and sex); histological subtypes of CMM (malignant, superficial spreading, nodular, lentigo maligna, Spitzoid or other); tumor site (lower limbs, upper limbs, head, hands/feet and trunk); Breslow thickness ($\leq 0.75$, 0.76–1.50, 1.51–3.99, $\geq 4.00$ mm); Clark level (I, II, III, IV, and V); CMM growth phase (radial versus vertical); ulceration (absent versus present); CMM regression (absent versus present); tumor-infiltrating lymphocytes (TIL) (absent versus present); mitotic count (number of mitoses per $mm^2$); T, N, and M AJCC (American Joint Committee on Cancer) stages at diagnosis (8[th] edition); and the number of positive lymph nodes after sentinel lymph node biopsy (SLNB).

## Statistical methods

Descriptive statistics were obtained to represent categorical variables as frequencies and proportions, whereas continuous numerical variables were summarized using means, standard deviations (SDs), medians, and minimum–maximum intervals.

A bivariate analysis was conducted; specifically, the Chi-squared test was used to evaluate the different distribution of qualitative variables by mitosis classes, while the difference in the mean value of quantitative variables by mitosis classes was verified through an ANOVA test.

Survival was firstly analyzed by plotting Kaplan–Meier short-term (3-year) overall survival (OS) probability curves, grouping the sample according to the following mitotic classes: 0, 1, 2, 3–5, and $\geq 6$ mitoses per $mm^2$.

In addition, the Cox proportional hazard model was used to evaluate the relationship between mitotic classes and 3-year OS in both univariate and multivariable models, adjusting for all variables associated with the mitotic count in bivariate analysis. However, the variables that define the T value for TNM staging were not added to the model to prevent over-adjustment. Various regressions were conducted based on different mitotic count groupings or cut-offs (creating a binary variable varying a single cut-off point from 1 to 10 mitoses per $mm^2$). The trend in significance of the Cox hazard ratio with the number of mitoses defining the cut-off was also evaluated with linear regression. Finally, the same analyses were repeated separately on the subpopulation subsets with T1, (T1a and T1B), T2, T3, and T4.

The results were considered statistically significant when $p < 0.05$. All data analyses were conducted in R 4.2.2 (RStudio, Inc., Boston, Massachusetts).

## Ethics approval and consent to participate

The study adheres to the Declaration of Helsinki and Resolution No 9/2016 of the Italian Data Protection Authority, with the latter also confirming the permissibility of processing personal data for medical, biomedical, and epidemiological research, as well as the permissibility of using data concerning the status of people's health in aggregate form in scientific studies. To protect privacy and anonymity, the Veneto Regional Authority removes all direct identifiers and replaces them with a code number in all datasets, while retaining the ability to link data

from different administrative databases. In this case, according to Resolution No 9/2016 of the Italian Data Protection Authority, written consent from patients is not required.

Ethical approval for the study was obtained from the Veneto Oncological Institute's Ethics Committee (No 52/2016).

## Results

Analyses were conducted on a total of 2,255 patients with primary melanoma; 46% were male. Table 1 depicts the clinical-anatomopathological characteristics of the sample. The mean age was 59.39 years (SD ± 16.06, min-max range 15–101). Half of the melanomas were located in the trunk; 76% of these were classified as stage I according to the AJCC TNM criteria (8th), and around 50% had no mitoses. The overall median follow-up time was 3.82 years (min-max 0.01–6.00). Stratifying by T value gave 3.95, 3.75, 3.63, and 3.11 years respectively for T1, T2, T3, and T4 subjects.

The bivariate analysis in Table 2 highlights statistically significant differences in many clinical and anatomopathological variables, including age, site, histological subtype, TNM stage, growth type, Breslow and Clark categories, and the presence of ulceration, tumor regression, TILs, or positive SLNB by mitotic group.

Survival analysis with Kaplan–Meier curves (Fig 1) revealed significant differences in OS based on the number of mitoses per $mm^2$ (Log-rank test $p < 0.0001$). Indeed, Cox regression confirmed that even variables not currently included in staging system—including CMM histology, site, and mitotic rate (MR)—remain associated with prognosis even after adjusting for TNM (Table 3). However, stratifying by T value, it was only possible to conclude that having a MR $\geq 6$ represents a significant risk factor for OS for subset T2 (hazard ratio (HR) = 4.20, $p = 0.010$).

Adopting single cut-off definitions for the number of mitoses, the multivariable Cox regression (Table 4) demonstrates that, for the entire sample, HR estimates were statistically significant only when the MR is $\geq 5$ or $\geq 6$ mitoses per $mm^2$. Subgroup analysis revealed that for T2 melanomas, the MR was statistically associated with OS when defined with cut-off points $\geq 5$ to $\geq 10$ mitoses. Conversely, in T1 patients, the MR only seems to approach statistical significance for cut-off points $\geq 7$, with significantly increasing HR ($p$ trend $< 0.0001$). Similar results were obtained in T1A and T1B subpopulations (data not shown). Survival analysis for T3 and T4 subsets did not reveal any significant prognostic feature of MR.

## Discussion

The results of this study indicate that the mitotic rate is associated with other prognostic factors for survival, including TNM-stage variables (e.g., tumor thickness, ulceration, lymph node status, and metastasis presence) and those not included in the TNM stage (e.g., age, site of tumor, type of morphology, growth pattern and TIL). Moreover, this study demonstrated that, even after adjusting for these prognostic factors, increased mitoses per $mm^2$ are associated with a higher risk of mortality, particularly in T2 patients.

As stated by earlier research [20], our findings demonstrated that prognostic factors are often interrelated. In particular, we observe that patients with a high mitotic rate are more likely to be older, male, have a melanoma lesion with a vertical growth pattern, nodular histological subtype, ulceration, greater tumor thickness, and an advanced stage. Also, previous observations highlighted that thick and ulcerated melanomas are mitotically active [21,22], which is compatible with the findings of our study. Moreover, a previous study [23] found that high-mitotic-rate primary cutaneous melanoma was associated with aggressive histologic features and an atypical clinical presentation, and that higher mitotic rates were more prevalent

**Table 1. Main demographic and clinical-pathological characteristics of the sample.**

| | Value | % (N = 2255) |
|---|---|---|
| **Age (at diagnosis), in years** | | |
| Mean | 59.39 | |
| Median | 59 | |
| SD | 16.06 | |
| Min-Max | 15–101 | |
| **Sex** | | |
| Male | 1,046 | 46.39 |
| Female | 1,209 | 53.61 |
| **Year of diagnosis** | | |
| 2015 | 1,050 | 46.56 |
| 2017 | 1,205 | 53.44 |
| **Primary site** | | |
| Trunk | 1,145 | 50.78 |
| Lower limb | 439 | 19.47 |
| Upper limb | 314 | 13.92 |
| Head | 242 | 10.73 |
| Hands/feet | 99 | 4.39 |
| Unknown | 16 | 0.71 |
| **M. Histology Subtype** | | |
| Superficial spreading | 1,677 | 74.37 |
| Nodular | 306 | 13.57 |
| Malignant (NOS) | 115 | 5.10 |
| Spitzoid | 55 | 2.44 |
| Lentigo maligna | 48 | 2.13 |
| Other | 54 | 2.39 |
| **Stage T** | | |
| T1 | 1,460 | 64.75 |
| T2 | 332 | 14.72 |
| T3 | 249 | 11.04 |
| T4 | 211 | 9.36 |
| Unknown | 3 | 0.13 |
| **Stage N** | | |
| N0 | 1,993 | 88.38 |
| N1 | 155 | 6.87 |
| N2 | 53 | 2.35 |
| N3 | 46 | 2.04 |
| Unknown | 8 | 0.35 |
| **Stage M** | | |
| M0 | 2,209 | 97.96 |
| M1 | 33 | 1.46 |
| Unknown | 13 | 0.58 |
| **TNM stage (AJCC 8th edition)** | | |
| I | 1,653 | 73.30 |
| II | 327 | 14.50 |
| III | 226 | 10.02 |
| IV | 33 | 1.46 |
| Unknown | 16 | 0.71 |

*(Continued)*

**Table 1.** (Continued)

| | Value | % (N = 2255) |
|---|---|---|
| **Growth pattern** | | |
| Vertical | 1,396 | 61.91 |
| Radial | 469 | 20.80 |
| Unknown | 390 | 17.29 |
| **Breslow thickness, in mm** | | |
| < 0.75 | 1,219 | 54.06 |
| 0.76–1.50 | 466 | 20.67 |
| 1.51–3.99 | 341 | 15.12 |
| $\geq 4$ | 225 | 9.98 |
| Unknown | 4 | 0.18 |
| **Clark level** | | |
| I | 5 | 0.22 |
| II | 596 | 26.43 |
| III | 774 | 34.32 |
| IV | 602 | 26.70 |
| V | 72 | 3.19 |
| Unknown | 206 | 9.14 |
| **Ulceration** | | |
| Absent | 1,831 | 81.20 |
| Present | 399 | 17.69 |
| Unknown | 25 | 1.11 |
| **Tumor regression** | | |
| Absent | 1,061 | 47.05 |
| Present | 639 | 28.34 |
| Unknown | 555 | 24.61 |
| **TILs** | | |
| Present | 1,647 | 73.04 |
| Absent | 465 | 20.62 |
| Unknown | 143 | 6.34 |
| **Positive SLNB** | | |
| 0 | 2,068 | 91.71 |
| 1 | 153 | 6.78 |
| >1 | 34 | 1.51 |
| **Mitoses per mm$^2$** | | |
| 0 | 1,126 | 49.93 |
| 1 | 330 | 14.63 |
| 2 | 202 | 8.96 |
| 3–5 | 259 | 11.49 |
| 6–9 | 150 | 6.65 |
| $\geq 10$ | 188 | 8.34 |
| **Follow-up, in years** | | |
| Mean | 4.16 | |
| Median | 3.82 | |
| SD | 1.29 | |
| Min-Max | 0.01–6.00 | |

Abbreviations: SD, Standard Deviations; NOS, Not Otherwise Specified; AJCC, American Joint Committee on Cancer; TIL, Tumor-Infiltrating Lymphocytes; SLNB, Sentinel Lymph Node Biopsy.

**Table 2. Demographic and clinical-pathological differences by number of mitoses per mm².**

| | Total (N = 2255) | Mitoses per mm² | | | | | | Test group p |
|---|---|---|---|---|---|---|---|---|
| | | 0 (N = 1126) | 1 (N = 330) | 2 (N = 202) | 3–5 (N = 259) | 6–9 (N = 150) | ≥ 10 (N = 188) | |
| **Age (at diagnosis), in years** | | | | | | | | < 0.0001 |
| Mean | 59.39 | 57.10 | 58.10 | 57.30 | 61.60 | 65.50 | 69.90 | |
| Median | 59 | 57 | 58 | 57 | 62 | 68 | 72 | |
| Min-Max | 15–101 | 16–94 | 15–94 | 18–93 | 18–97 | 23–95 | 23–101 | |
| **Sex** | | | | | | | | 0.08885 |
| Male | 1,046 (46.39) | 584 (51.87) | 172 (52.12) | 110 (54.46) | 141 (54.44) | 97 (64.67) | 105 (55.85) | |
| Female | 1,209 (53.61) | 542 (48.13) | 158 (47.88) | 92 (45.54) | 118 (45.56) | 53 (35.33) | 83 (44.15) | |
| **Year of diagnosis, n (%)** | | | | | | | | 0.06974 |
| 2015 | 1,050 (46.56) | 552 (49.02) | 156 (47.27) | 90 (44.55) | 113 (43.63) | 55 (36.67) | 84 (44.68) | |
| 2017 | 1,205 (53.44) | 574 (50.98) | 174 (52.73) | 112 (55.45) | 146 (56.37) | 95 (63.33) | 104 (55.32) | |
| **Primary site, n (%)** | | | | | | | | < 0.0001 |
| Trunk | 1,145 (50.78) | 636 (56.48) | 158 (47.88) | 93 (46.04) | 123 (47.49) | 63 (42.00) | 72 (38.30) | |
| Lower limb | 439 (19.47) | 206 (18.29) | 72 (21.82) | 48 (23.76) | 57 (22.01) | 27 (18.00) | 29 (15.43) | |
| Upper limb | 314 (13.92) | 136 (12.08) | 45 (13.64) | 33 (16.34) | 36 (13.90) | 29 (19.33) | 35 (18.62) | |
| Head | 242 (10.73) | 97 (8.61) | 41 (12.42) | 20 (9.90) | 26 (10.04) | 23 (15.33) | 35 (18.62) | |
| Hands/feet | 99 (4.39) | 42 (3.73) | 13 (3.94) | 8 (3.96) | 13 (5.02) | 7 (4.67) | 16 (8.51) | |
| Unknown | 16 (0.71) | 9 (0.80) | 1 (0.30) | 0 (0.00) | 4 (1.54) | 1 (0.67) | 1 (0.53) | |
| **M. Histology Subtype, n (%)** | | | | | | | | < 0.0001 |
| Superficial spreading | 1,677 (74.37) | 1,003 (89.08) | 271 (82.12) | 140 (69.31) | 141 (54.44) | 65 (43.33) | 57 (30.32) | |
| Nodular | 306 (13.57) | 7 (0.62) | 16 (4.85) | 33 (16.34) | 81 (31.27) | 63 (42.00) | 106 (56.38) | |
| Malignant (NOS) | 115 (5.10) | 27 (2.4) | 21 (6.36) | 19 (9.41) | 19 (7.34) | 13 (8.67) | 16 (8.51) | |
| Lentigo maligna | 55 (2.44) | 35 (3.11) | 5 (1.52) | 2 (0.99) | 1 (0.39) | 2 (1.33) | 3 (1.60) | |
| Spitzoid | 48 (2.13) | 25 (2.22) | 10 (3.03) | 5 (2.48) | 11 (4.25) | 3 (2.00) | 1 (0.53) | |
| Other | 54 (2.39) | 29 (2.58) | 7 (2.12) | 3 (1.49) | 6 (2.32) | 4 (2.67) | 5 (2.66) | |
| **Stage T, n (%)** | | | | | | | | < 0.0001 |
| T1 | 1,460 (64.75) | 1,079 (95.83) | 241 (73.03) | 90 (44.55) | 40 (15.44) | 9 (6.00) | 1 (0.53) | |
| T2 | 332 (14.72) | 38 (3.37) | 71 (21.52) | 71 (35.15) | 93 (35.91) | 35 (23.33) | 24 (12.77) | |
| T3 | 249 (11.04) | 6 (0.53) | 14 (4.24) | 31 (15.35) | 77 (29.73) | 58 (38.67) | 63 (33.51) | |
| T4 | 211 (9.36) | 2 (0.18) | 3 (0.91) | 10 (4.95) | 48 (18.53) | 48 (32.00) | 100 (53.19) | |
| Unknown | 3 (0.13) | 1 (0.09) | 1 (0.30) | 0 (0.00) | 1 (0.39) | 0 (0.00) | 0 (0.00) | |
| **Stage N, n (%)** | | | | | | | | < 0.0001 |
| N0 | 1,993 (88.38) | 1,120 (99.47) | 304 (92.12) | 173 (85.64) | 184 (71.04) | 103 (68.67) | 109 (57.98) | |
| N1 | 155 (6.87) | 5 (0.44) | 18 (5.45) | 17 (8.42) | 46 (17.76) | 26 (17.33) | 43 (22.87) | |
| N2 | 53 (2.35) | 1 (0.09) | 6 (1.82) | 7 (3.47) | 18 (6.95) | 9 (6.00) | 12 (6.38) | |
| N3 | 46 (2.04) | 0 (0.00) | 1 (0.30) | 5 (2,48) | 8 (3.09) | 10 (6.67) | 22 (11.70) | |
| Unknown | 8 (0.35) | 0 (0.00) | 1 (0.30) | 0 (0.00) | 3 (1.16) | 2 (1.33) | 2 (1.06) | |
| **Stage M, n (%)** | | | | | | | | < 0.0001 |
| M0 | 2,209 (97.96) | 1,124 (99.82) | 326 (98.79) | 198 (98.02) | 250 (96.53) | 141 (94.00) | 170 (90.43) | |
| M1 | 33 (1.46) | 2 (0.18) | 2 (0.61) | 3 (1.49) | 6 (2.32) | 6 (4.00) | 14 (7.45) | |
| Unknown | 13 (0.58) | 0 (0.00) | 2 (0.60) | 1 (0.50) | 3 (1.16) | 3 (2.00) | 4 (2.13) | |
| **TNM stage, n (%)** | | | | | | | | < 0.0001 |
| I | 1,653 (73.30) | 1,106 (98.22) | 286 (86.67) | 134 (66.34) | 84 (32.43) | 30 (20.00) | 13 (6.91) | |
| II | 327 (14.50) | 11 (0.98) | 17 (5.15) | 38 (18.81) | 97 (37.45) | 72 (48.00) | 92 (48.94) | |
| III | 226 (10.02) | 6 (0.53) | 22 (6.67) | 26 (12.87) | 68 (26.25) | 39 (26.00) | 65 (34.57) | |
| IV | 33 (1.46) | 2 (0.18) | 2 (0.61) | 3 (1.49) | 6 (2.32) | 6 (4.00) | 14 (7.45) | |

(*Continued*)

**Table 2.** (Continued)

| | Total (N = 2255) | Mitoses per mm$^2$ | | | | | | Test group $p$ |
|---|---|---|---|---|---|---|---|---|
| | | 0 (N = 1126) | 1 (N = 330) | 2 (N = 202) | 3–5 (N = 259) | 6–9 (N = 150) | ≥ 10 (N = 188) | |
| Unknown | 16 (0.71) | 1 (0.09) | 3 (0.90) | 1 (0.50) | 4 (1.54) | 3 (2.00) | 4 (2.13) | |
| **Growth pattern, n (%)** | | | | | | | | < 0.0001 |
| Vertical | 1,396 (61.91) | 509 (45.20) | 253 (76.67) | 167 (82.67) | 206 (79.54) | 124 (82.67) | 137 (72.87) | |
| Radial | 469 (20.80) | 422 (37.48) | 32 (9.70) | 6 (2.97) | 5 (1.93) | 2 (1.33) | 2 (1.06) | |
| Unknown | 390 (17.29) | 195 (17.32) | 45 (13.64) | 29 (14.36) | 48 (18.53) | 24 (16.00) | 49 (26.07) | |
| **Breslow thickness, in mm, n (%)** | | | | | | | | < 0.0001 |
| < 0.75 | 1,219 (54.06) | 992 (88.10) | 163 (49.39) | 44 (21.78) | 16 (6.18) | 4 (2.67) | 0 (0.00) | |
| 0.76–1.50 | 466 (20.67) | 122 (10.83) | 137 (41.52) | 93 (46.04) | 76 (29.34) | 28 (18.67) | 10 (5.32) | |
| 1.51–3.99 | 341 (15.12) | 8 (0.71) | 26 (7.88) | 54 (26.73) | 114 (44.02) | 65 (43.33) | 74 (39.36) | |
| ≥ 4 | 225 (9.98) | 2 (0.18) | 3 (0.91) | 11 (5.45) | 52 (20.08) | 53 (35.33) | 104 (55.32) | |
| Unknown | 4 (0.18) | 2 (0.18) | 1 (0.30) | 0 (0.00) | 1 (0.39) | 0 (0.00) | 0 (0.00) | |
| **Clark level, n (%)** | | | | | | | | < 0.0001 |
| I | 5 (0.22) | 4 (0.36) | 1 (0.30) | 0 (0.00) | 0 (0.00) | 0 (0.00) | 0 (0.00) | |
| II | 596 (26.43) | 538 (47.78) | 43 (13.03) | 9 (4.46) | 5 (1.93) | 1 (0.67) | 0 (0.00) | |
| III | 774 (34.32) | 430 (38.19) | 156 (47.27) | 75 (37.13) | 67 (25.87) | 29 (19.33) | 17 (9.04) | |
| IV | 602 (26.70) | 74 (6.57) | 91 (27.58) | 93 (46.04) | 143 (55.21) | 91 (60.67) | 110 (58.51) | |
| V | 72 (3.19) | 1 (0.09) | 1 (0.30) | 5 (2.48) | 16 (6.18) | 19 (12.67) | 30 (15.96) | |
| Unknown | 206 (9.14) | 79 (7.02) | 38 (11.52) | 20 (9.90) | 28 (10.82) | 10 (6.67) | 31 (16.49) | |
| **Ulceration, n (%)** | | | | | | | | < 0.0001 |
| Absent | 1,831 (81.20) | 1,092 (96.98) | 293 (88.79) | 163 (80.69) | 156 (60.23) | 76 (50.67) | 51 (27.13) | |
| Present | 399 (17.69) | 22 (1.95) | 29 (8.79) | 36 (17.82) | 102 (39.38) | 73 (48.67) | 137 (72.87) | |
| Unknown | 25 (1.11) | 12 (1.07) | 8 (2.42) | 3 (1.49) | 1 (0.39) | 1 (0.67) | 0 (0.00) | |
| **Tumor regression, n (%)** | | | | | | | | < 0.0001 |
| Absent | 1,061 (47.05) | 464 (41.21) | 160 (48.48) | 98 (48.51) | 138 (53.28) | 94 (62.67) | 107 (56.91) | |
| Present | 639 (28.34) | 418 (37.12) | 103 (31.21) | 38 (18.81) | 52 (20.08) | 18 (12.00) | 10 (5.32) | |
| Unknown | 555 (24.61) | 244 (21.67) | 67 (20.30) | 66 (32.67) | 69 (24.64) | 38 (25.33) | 71 (37.77) | |
| **TILs, n (%)** | | | | | | | | 0.001974 |
| Present | 1,647 (73.04) | 827 (73.45) | 247 (74.85) | 153 (75.74) | 194 (74.90) | 102 (68.00) | 124 (65.96) | |
| Absent | 465 (20.62) | 200 (17.76) | 67 (20.30) | 41 (20.30) | 57 (22.01) | 41 (27.33) | 59 (31.38) | |
| Unknown | 143 (6.34) | 99 (8.79) | 16 (4.85) | 8 (3.96) | 8 (3.09) | 7 (4.67) | 5 (2.66) | |
| **Positive SLNB, n (%)** | | | | | | | | < 0.0001 |
| 0 | 2,068 (91.71) | 1,121 (99.56) | 308 (93.33) | 177 (87.62) | 201 (77.61) | 120 (80.00) | 141 (75.00) | |
| 1 | 153 (6.78) | 5 (0.44) | 18 (5.45) | 20 (9.90) | 50 (19.31) | 26 (17.33) | 34 (18.09) | |
| >1 | 34 (1.51) | 0 (0.00) | 4 (1.21) | 5 (2.48) | 8 (3.09) | 4 (2.67) | 13 (6.91) | |
| **Deceased in 3 years, n (%)** | | | | | | | | < 0.0001 |
| No | 2,069 (91.75) | 1,097 (97.42) | 318 (96.36) | 190 (94.06) | 228 (88.03) | 112 (74.67) | 124 (65.96) | |
| Yes | 186 (8.25) | 29 (2.58) | 12 (3.64) | 12 (5.94) | 31 (11.97) | 38 (25.33) | 64 (34.04) | |

Abbreviations: NOS, Not Otherwise Specified; TIL, Tumor-Infiltrating Lymphocytes; SLNB, Sentinel Lymph Node Biopsy.

in older, male patients with rapidly growing primary melanoma that was more commonly located on the head and neck. Melanomas with very high mitotic activity (≥ 10 mitoses/mm$^2$) were predominantly thick and ulcerated nodular tumor subtypes. In contrast, the superficial spreading melanoma subtype and regression characteristics were found to be typical of lesions with sparse mitotic activity. These melanomas were significantly thinner than their

**Table 3. Unadjusted and adjusted Cox regression hazard ratios (HR) and *p*-value estimates count variable.**

| | | All subjects (N = 2255) | | | | Only T1 subjects (N = 1460) | | | | Only T2 subjects (N = 332) | | | | Only T3 subjects (N = 249) | | | | Only T4 subjects (N = 211) | | | |
|---|---|---|---|---|---|---|---|---|---|---|---|---|---|---|---|---|---|---|---|---|---|
| | | Univariate reg. | | Multiv. reg.* | | Univariate reg. | | Multiv. reg.* | | Univariate reg. | | Multiv. reg.* | | Univariate reg. | | Multiv. reg.* | | Univariate reg. | | Multiv. reg.* | |
| | | HR | *p* | HR | *p* | HR | *p* | HR | *p* | HR | *p* | HR | *p* | HR | *p* | HR | *p* | HR | *p* | HR | *p* |
| Mitoses per mm² (ref. 0–2) | 3–5 | 3.90 | <0.0001 | 1.09 | 0.774 | 0.00 | 0.998 | 0.00 | 0.996 | 0.63 | 0.482 | 0.75 | 0.711 | 0.98 | 0.961 | 1.29 | 0.666 | 1.40 | 0.544 | 1.44 | 0.542 |
| | ≥6 | 11.07 | <0.0001 | 1.88 | 0.019 | 4.50 | 0.138 | 2.74 | 0.329 | 1698.00 | 0.343 | 4.20 | 0.010 | 1.16 | 0.733 | 0.95 | 0.925 | 2.43 | 0.084 | 2.42 | 0.116 |
| Age | | 1.08 | <0.0001 | 1.08 | <0.0001 | 1.14 | <0.0001 | 1.16 | <0.0001 | 1.09 | <0.0001 | 1.11 | <0.0001 | 1.06 | <0.0001 | 1.09 | <0.0001 | 1.03 | <0.0001 | 1.04 | 0.0002 |
| Sex (ref. Female) | Male | 1.45 | 0.015 | 1.27 | 0.215 | 2.31 | 0.026 | 1.65 | 0.259 | 2.75 | 0.077 | 1.50 | 0.490 | 1.18 | 0.625 | 1.62 | 0.334 | 0.78 | 0.230 | 0.90 | 0.698 |
| Year of diagnosis (ref. 2015) | 2017 | 1.14 | 0.382 | 0.78 | 0.154 | 1794.00 | 0.103 | 0.87 | 0.743 | 0.50 | 0.154 | 0.26 | 0.013 | 1.18 | 0.614 | 0.86 | 0.726 | 0.74 | 0.146 | 0.59 | 0.048 |
| Primary site (ref. Lower limb) | Upper limb | 0.67 | 0.164 | 0.41 | 0.010 | 1.24 | 0.721 | 0.34 | 0.148 | 0.47 | 0.511 | 0.32 | 0.266 | 0.15 | 0.076 | 0.19 | 0.146 | 0.57 | 0.147 | 0.70 | 0.434 |
| | Head | 1.66 | 0.033 | 0.85 | 0.589 | 2.42 | 0.126 | 0.50 | 0.174 | 0.00 | 0.997 | 0.00 | 0.989 | 1.41 | 0.518 | 1.17 | 0.831 | 0.86 | 0.629 | 0.71 | 0.387 |
| | Hands/feet | 2.53 | 0.0008 | 1.03 | 0.926 | 0.00 | 0.996 | 0.00 | 0.997 | 3.61 | 0.116 | 3.01 | 0.149 | 1.95 | 0.302 | 6.57 | 0.014 | 1.01 | 0.977 | 0.69 | 0.400 |
| | Trunk | 0.76 | 0.164 | 0.78 | 0.318 | 0.95 | 0.907 | 0.59 | 0.206 | 1.23 | 0.754 | 0.80 | 0.689 | 1.16 | 0.750 | 0.99 | 0.983 | 0.52 | 0.019 | 0.65 | 0.235 |
| M. Histology Subtype (ref. Superficial spreading) | Malignant (NOS) | 1.20 | 0.647 | 0.87 | 0.801 | 0.00 | 0.997 | 0.00 | 0.993 | 0.91 | 0.900 | 0.00 | 0.994 | 0.897 | 0.885 | 6.45 | 0.044 | 0.50 | 0.256 | 0.30 | 0.261 |
| | Nodular | 5.91 | <0.0001 | 0.96 | 0.849 | 0.00 | 0.998 | 0.00 | 0.998 | 0.00 | 0.997 | 0.00 | 0.997 | 1.25 | 0.517 | 1.20 | 0.668 | 0.85 | 0.495 | 0.80 | 0.457 |
| | Lentigo maligna | 1.65 | 0.329 | 0.35 | 0.171 | 2.19 | 0.283 | 0.67 | 0.692 | 0.00 | 0.999 | - | 0.998 | 4.18 | 0.998 | 0.00 | 0.998 | 3.47 | 0.093 | 0.29 | 0.282 |
| | Spitzoid | 0.35 | 0.292 | 1.16 | 0.884 | 0.00 | 0.997 | 0.00 | 0.997 | 0.00 | 0.999 | 0.00 | 0.999 | 1.32 | 0.787 | 3.96 | 0.223 | 1.60 | 0.994 | 0.00 | 0.996 |
| | Other | 2.71 | 0.011 | 0.73 | 0.555 | 0.00 | 0.998 | 0.00 | 0.997 | 1.56 | 0.669 | 0.42 | 0.400 | 0.00 | 0.997 | 0.00 | 0.998 | 1.34 | 0.556 | 0.99 | 0.991 |
| TILs (ref. Absent) | Present | 0.54 | 0.0001 | 1.10 | 0.616 | 0.65 | 0.302 | 1.11 | 0.822 | 1.28 | 0.696 | 0.63 | 0.547 | 0.99 | 0.987 | 1.24 | 0.644 | 0.74 | 0.165 | 1.12 | 0.693 |
| Growth pattern (ref. Radial) | Vertical | 3.52 | <0.0001 | 0.80 | 0.515 | 1.08 | 0.837 | 0.80 | 0.616 | 0.547 | 0.560 | 0.20 | 0.122 | 0.29 | 0.218 | 0.24 | 0.244 | 0.30 | 0.094 | 0.76 | 0.728 |
| TNM stage (ref. I) | II | 8.64 | <0.0001 | 3.55 | <0.0001 | - | - | 1.00 | 0.999 | 1.44 | 0.589 | 2.22 | 0.222 | 0.09 | <0.0001 | 0.06 | <0.0001 | 0.33 | 0.0004 | 0.18 | 0.0001 |
| | III | 10.90 | <0.0001 | 5.56 | <0.0001 | 0.00 | 0.996 | 0.00 | 0.994 | 2.06 | 0.239 | 4.19 | 0.016 | 0.13 | <0.0001 | 0.08 | 0.005 | 0.47 | 0.016 | 0.30 | 0.003 |
| | IV | 56.11 | <0.0001 | 34.83 | <0.0001 | 1437.50 | <0.0001 | 67.92 | 0.003 | 24.92 | 0.002 | - | 0.988 | - | - | - | - | - | - | - | - |

Abbreviations: NOS, Not Otherwise Specified; TIL, Tumor-Infiltrating Lymphocytes.

* Adjusting by age, sex, site, cohort year, TILs, growth type, histology, and TNM stage.

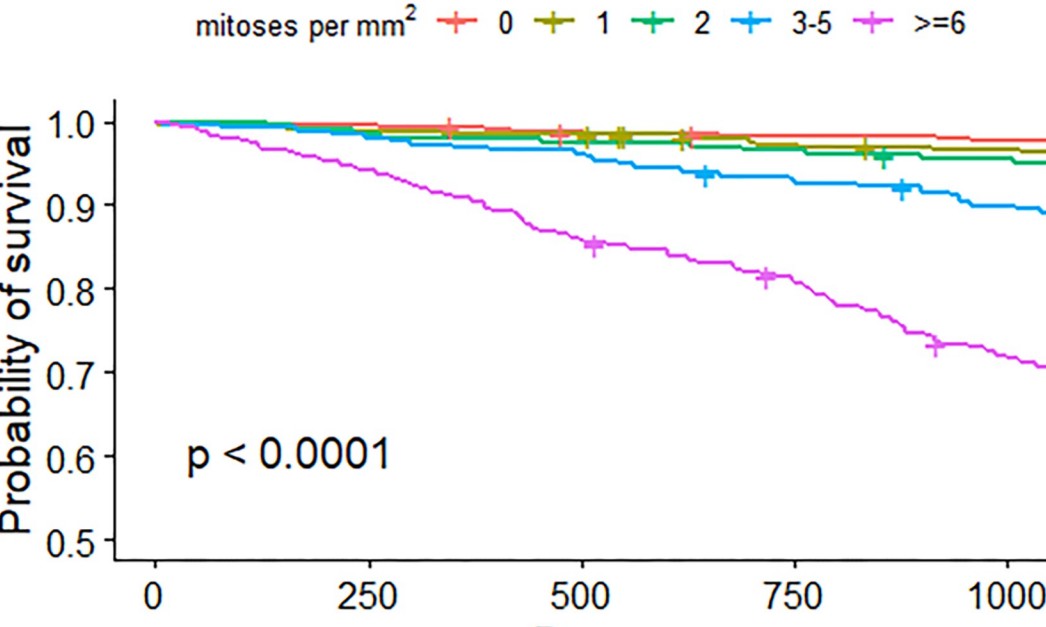

**Fig 1. Kaplan–Meier survival curves in CMM subjects by number of mitoses per mm$^2$.**

counterparts with greater mitotic activity. Similarly, our study demonstrated that more than half of the melanomas with a very high mitotic activity ($\geq$ 10 mitoses/mm$^2$) were nodular melanomas, whereas less than 1% of tumors with 0 mitoses/mm$^2$ were nodular, and the frequency

**Table 4. Unadjusted and adjusted Cox regression hazard ratios (HR) and *p*-value estimates for different representations of the mitotic count variable.**

| Mitoses per mm$^2$ | | All subjects (N = 2255) | | | | Only T1 subjects (N = 1460) | | | | Only T2 subjects (N = 332) | | | | Only T3 subjects (N = 249) | | | | Only T4 subjects (N = 211) | | | |
|---|---|---|---|---|---|---|---|---|---|---|---|---|---|---|---|---|---|---|---|---|---|
| | | Univariate reg. | | Multiv. reg.* | | Univariate reg. | | Multiv. reg.* | | Univariate reg. | | Multiv. reg.* | | Univariate reg. | | Multiv. reg.* | | Univariate reg. | | Multiv. reg.* | |
| | | HR | *p* | HR | *p* | HR | *p* | HR | *p* | HR | *p* | HR | *p* | HR | *p* | HR | *p* | HR | *p* | HR | *p* |
| Single threshold (reference <) | $\geq$1 | 5.75 | <**0.0001** | 1.34 | 0.388 | 0.61 | 0.265 | 0.68 | 0.441 | 2.10 | 0.473 | 1.85 | 0.553 | - | 0.996 | - | 0.997 | - | 0.995 | - | 0.996 |
| | $\geq$2 | 6.99 | <**0.0001** | 1.43 | 0.239 | 0.58 | 0.461 | 0.89 | 0.880 | 0.88 | 0.805 | 1.28 | 0.676 | 1.50 | 0.576 | 1.77 | 0.466 | 2.53 | 0.355 | 1.38 | 0.752 |
| | $\geq$3 | 7.74 | <**0.0001** | 1.55 | 0.086 | 0.85 | 0.877 | 1.36 | 0.763 | 1.03 | 0.946 | 1.95 | 0.211 | 1.09 | 0.839 | 1.06 | 0.904 | 2.15 | 0.133 | 1.99 | 0.211 |
| | $\geq$4 | 7.86 | <**0.0001** | 1.51 | 0.074 | 1.92 | 0.520 | 1.72 | 0.597 | 1.19 | 0.732 | 2.57 | 0.081 | 1.03 | 0.922 | 0.75 | 0.497 | 2.01 | 0.058 | 1.63 | 0.256 |
| | $\geq$5 | 7.72 | <**0.0001** | 1.58 | **0.032** | 3.01 | 0.278 | 2.65 | 0.344 | 1.51 | 0.438 | 4.01 | **0.013** | 1.01 | 0.978 | 0.76 | 0.499 | 1.85 | 0.029 | 1.63 | 0.151 |
| | $\geq$6 | 8.03 | <**0.0001** | 1.79 | **0.005** | 4.63 | 0.131 | 2.73 | 0.331 | 1.95 | 0.209 | 4.49 | **0.007** | 1.18 | 0.613 | 0.83 | 0.665 | 1.87 | 0.013 | 1.85 | 0.060 |
| | $\geq$7 | 6.94 | <**0.0001** | 1.37 | 0.127 | 9.98 | **0.023** | 5.68 | 0.098 | 3.34 | **0.023** | 5.42 | **0.002** | 0.86 | 0.657 | 0.72 | 0.489 | 1.38 | 0.134 | 1.09 | 0.829 |
| | $\geq$8 | 7.35 | <**0.0001** | 1.41 | 0.100 | 12.96 | **0.012** | 6.04 | 0.087 | 2.95 | 0.058 | 4.14 | **0.016** | 1.14 | 0.691 | 0.97 | 0.950 | 1.41 | 0.101 | 1.13 | 0.697 |
| | $\geq$9 | 6.60 | <**0.0001** | 1.26 | 0.271 | 32.08 | **0.0006** | 6.04 | 0.087 | 3.59 | **0.025** | 4.18 | **0.016** | 1.14 | 0.707 | 0.99 | 0.978 | 1.16 | 0.463 | 0.88 | 0.670 |
| | $\geq$10 | 6.81 | <**0.0001** | 1.21 | 0.375 | 126.12 | <**0.0001** | 6.04 | 0.087 | 4.11 | **0.013** | 4.45 | **0.012** | 1.21 | 0.589 | 1.06 | 0.911 | 1.15 | 0.494 | 0.76 | 0.388 |
| *p* HR trend | | 0.863 | | 0.317 | | **0.031** | | <**0.0001** | | **0.002** | | **0.003** | | 0.525 | | 0.319 | | <**0.0001** | | **0.014** | |

* Adjusting by age, sex, site, cohort year, TILs, growth type, histology, and TNM stage.

of superficial melanoma was lower in the high mitotic activity rate group (89% in the group with 0 mitosis/mm$^2$ vs. 30% in the group with $\geq$ 10 mitosis/mm$^2$). Nodular melanoma tended to be more closely linked to solar keratosis than superficial spreading melanoma [23]. Overall, the results of our population-based study confirm the results of a previous single-center study by Katsambas *et al* [24], and define a high-risk profile for melanoma, a distinct phenotypic and histologic profile, associated with a high MR. The identification of this profile could help to develop a more precise and personalized approach for individual patients, at least in terms of both planning of staging and follow-up protocols.

Furthermore, the results of our analyses on overall short-term survival (OS) highlighted the fact that mitoses per mm$^2$, even after adjusting for these interrelated prognostic factors, are associated with higher mortality, particularly in T2 patients. However, although the previous AJCC classification [10] recognizes two tumor-associated factors for thin melanomas—mitotic rate (MR) and ulceration—in the latest edition (8[th]), the evaluation of the number of mitoses/ mm$^2$ is no longer used within the T1 category to differentiate pT1a from pT1b, leading to controversy over the viability of including the presence of mitoses as an essential variable for defining staging [11,25]. Although mitotic rate was removed as a T category criterion in the latest AJCC staging system, it remains a strong predictor of survival and continues to be documented in CMM pathology reports. Similarly, this count has been used to predict aggressive behavior in several neoplasms [5,26–28]. Our findings confirmed that the presence of mitosis has a direct prognostic value, showing that the survival prognosis in stage IA did not appear to be associated with the number of mitoses. For patients in stage T1B, the HR appeared to increase from 7 mitoses/mm$^2$, albeit not reaching statistical significance. However, there is a clear upward trend ($p$ = 0.003) in the hazard ratio for T2 patients, suggesting that independent risk factors play a significant role in the survival of this subgroup of patients. Concordantly, the SIAPEC-IAP 2022 Guidelines state that, although the mitotic index is no longer used as a prognostic criterion for the T1 category of primitive skin melanomas, it will likely be incorporated into the new models of personalized prognostics in the near future [29]. Other studies have sought to establish whether the presence of mitosis has a direct prognostic value [22,30,31]. A study by Thompson of 13,269 patients in the AJCC melanoma staging database showed that a high mitotic rate in a primary melanoma is associated with a lower survival probability and was the strongest prognostic factor after tumor thickness [28,31]. Recently, a retrospective analysis showed that the mitotic rate is the second strongest predictor of melanoma-specific survival after sentinel lymph node involvement [32], and a large series from the National Cancer Institute's Surveillance, Epidemiology, and End Results registry found mitotic tumor rate to be an independent predictor of survival for localized melanoma [11]. A recent review demonstrated that a single mitosis in thin melanomas did not increase the risk of sentinel lymph node positivity, although it was related to a significant decrease in survival rate [33]. A retrospective study comparing de novo melanomas and nevus-associated melanomas found that there are no prognostic differences when the mitotic rate is considered for one or more mitoses; however, there are prognostic differences when this criterion is considered for more than five mitoses, which raises the question of what should be the cut-off point for this variable [34]. Indeed, as discussed in [14], incorporating the MR into the staging system has proven difficult given the nonlinear nature of its impact on survival. In agreement with our work, the results of Kashani *et al* show that, for different values of T, the optimal cut-offs are not always the same. Further validation of the best thresholds would therefore seem to be necessary before reintroducing MR into the T category.

Overall, this work is consistent with the recent literature. Considering its strengths and limitations, as a population-based study (rather than center-specific), the risk of selection bias was minimized. We also believe that the observed cohorts are relevant in terms of sample size.

Nevertheless, its short follow-up duration means that future studies are needed to demonstrate the role of mitosis in long-term outcomes. We are aware that other prognostic variables were also not evaluated in our work, such as metastasis or treatments. Despite the fact that previous studies [35] identified the micro/macro metastatic pattern of sentinel lymph node invasion as a predictor of non-sentinel lymph node involvement and overall survival, we were unable to estimate its association as the data were not available in our registry. Moreover, the present analysis, which is based on real-world data, does not account for (potentially significant) new oncological therapies hoped to significantly alter survival. However, the Veneto Regional Oncology Network has produced a comprehensive clinical pathway detailing the clinical procedures to be applied in each step of the clinical management of melanoma patients, standardizing therapy and follow-up management for patients across the region according to the best clinical evidence [16]. Finally, being a population study, a further limitation of the present work is the underrepresentation of subjects with T3 or T4 melanoma at diagnosis. Low sample size may have hindered statistical significance and power in these subpopulations. In our region (Veneto, Italy), prevention campaigns over the past 30 years have increased population awareness about pigmented lesions and, together with early diagnosis, have indeed significantly reduced the incidence of advanced-stage melanomas.

On the other hand, we believe that the analysis of a large group of T1 and T2 patients, such as ours, allowed for interesting observations on MR value, particularly for T2 stage. The results of the present study support the potential proposal for MR-based personalized planning of pre-operative staging investigations and postsurgical follow-up protocols for T2N0M0 patients (IB and IIA stage), including the highly debated adjuvant therapy [16].

## Conclusions

In conclusion, these findings, along with the related existing studies, revealed the need to re-include the mitotic rate into the T category of AJCC staging classification, since it correlates with the prognosis as an independent factor, potentially enabling greater accuracy in the prediction of survival. In addition, the mitotic rate can be used to develop a personalized medicine approach in the treatment and follow-up of melanoma patients.

## Acknowledgments

We thank Stefano Guzzinati from the Veneto Tumor Registry, Azienda Zero (Padova, Italy), for his help in creating the public repository of our anonymized minimal data set.

## Author Contributions

**Conceptualization:** Alessandra Buja, Claudia Cozzolino.

**Data curation:** Claudia Cozzolino, Manuel Zorzi, Paolo Del Fiore, Saveria Tropea.

**Formal analysis:** Alessandra Buja, Claudia Cozzolino, Giovanna Boccuzzo.

**Investigation:** Alessandra Buja.

**Methodology:** Alessandra Buja.

**Project administration:** Alessandra Buja, Simone Mocellin.

**Software:** Claudia Cozzolino.

**Supervision:** Alessandra Buja, Massimo Rugge, Luigi dall'Olmo, Carlo Riccardo Rossi, Giovanna Boccuzzo, Simone Mocellin.

**Visualization:** Claudia Cozzolino.

**Writing – original draft:** Alessandra Buja, Francesca Dossi.

**Writing – review & editing:** Alessandra Buja, Massimo Rugge, Claudia Cozzolino, Francesca Dossi, Antonella Vecchiato, Giuseppe de Luca, Saveria Tropea, Luigi dall'Olmo.

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
