## [Decision Letter · Decision Letter 0]

25 Oct 2023

PONE-D-22-26758Could the mitotic count improve personalized prognosis in melanoma patients?

PLOS ONE

Dear Dr. Cozzolino,

Thank you for submitting your manuscript to PLOS ONE. After careful consideration, we feel that it has merit but does not fully meet PLOS ONE’s publication criteria as it currently stands. Therefore, we invite you to submit a revised version of the manuscript that addresses the points raised during the review process.

*Comments from PLOS Editorial Office: We note that one or more reviewers has recommended that you cite specific previously published works. As always, we recommend that you please review and evaluate the requested works to determine whether they are relevant and should be cited. It is not a requirement to cite these works. We appreciate your attention to this request.* Please review this paper and address the following questions/criticism to strengthen the data. Please submit your revised manuscript by Dec 09 2023 11:59PM. If you will need more time than this to complete your revisions, please reply to this message or contact the journal office at plosone@plos.org. Please include the following items when submitting your revised manuscript:

We look forward to receiving your revised manuscript.

Kind regards,

Habib Boukerche, PhD

Academic Editor

PLOS ONE

**Journal requirements:**

2. Our staff editors have determined that your manuscript is likely within the scope of our Personalised Medicine Call for Papers. This call for papers aims to showcase the breadth of research within personalized medicine and its impact on innovating the healthcare horizon.. Additional information can be found on our announcement page: Personalized Medicine Call for Paper- PLOS Collection

If you would like your manuscript to be considered for this collection, please let us know in your cover letter and we will ensure that your paper is treated as if you were responding to this call. Please note that being considered for the Collection does not require additional peer review beyond the journal’s standard process and will not delay the publication of your manuscript if it is accepted by PLOS ONE. If you would prefer to remove your manuscript from collection consideration, please specify this in the cover letter.

**Additional Editor Comments **:

The introduction has to be re-­‐structured, and the rationale for the study should be set clear. The manuscript should also be checked for types and grammars errors.

**Comments to the Author**

1. Is the manuscript technically sound, and do the data support the conclusions?

Reviewer #1: Yes

Reviewer #2: Yes

2. Has the statistical analysis been performed appropriately and rigorously? 

Reviewer #1: Yes

Reviewer #2: Yes

3. Have the authors made all data underlying the findings in their manuscript fully available?

Reviewer #1: Yes

Reviewer #2: Yes

4. Is the manuscript presented in an intelligible fashion and written in standard English?

Reviewer #1: Yes

Reviewer #2: Yes

5. Review Comments to the Author

***Reviewer #1***: This is an interesting study looking at the prognostic role of Mitoses in melanoma patients. This topic has been discussed in the past years , since it gave indication to SLN biopsy in the last AJCC classification (upstaging the T1a to T1b).

The study is well conducted and the results and discussion sound interesting. There authors made correction with SLNB status but not with the involvement of SLN (micro macro metastasis). Please have a look at Quaglino et al . Surg Oncol. 2011 Dec;20(4):259-64. Is there a particular reason not to look at this association ? Please discuss.

***Reviewer #2***: In this manuscript the authors examine the prognostic significance of mitotic rate in two cohorts of melanoma patients from the Veneto Cancer Registry. They show a correlation between increasing mitotic rate and various histopathologic factors in melanoma. They also show worse survival with increasing mitotic rate by Kaplan-Meier analysis. They investigate various cutoffs for mitotic rate in the entire cohorts, and in individual T categories. They conclude that mitotic rate does have prognostic significance, and may be used in individualized prognostic assessments of melanoma patients. Overall, the manuscript is useful in its reinforcement of the importance of mitotic rate. Attention to the following points is warranted to strengthen their conclusions, and also to address limitations in the available cohorts that precluded additional observations regarding the prognostic significance of mitotic rate.

1. A critical issue regarding assessment of mitotic rate, given the non-linear shape of its impact on survival, is the identification of optimal cut-points, both in the entire cohort of patients, as well as in specific T categories, as has been previously demonstrated (ref 14). To begin with, this should be discussed in greater detail in the Discussion section. While the authors examine this issue in some analyses, additional analyses are suggested. For example, in their K-M analysis of the entire cohort in Fig. 1, the authors examine different cutpoints for mitotic rate (0, 1, 2, 3-5; and >6). This analysis suggests the following “optimal” grouping or index for mitotic rate with differing survival: 0-2; 3-5; and > 6. It would be both interesting and important to analyze this further, both in univariate and multivariate analyses for the entire cohort. Also, given that the analysis of individual cutpoints only showed significant results in T2 patients, it would be interesting to examine this index in each T category to determine whether it would show a significant prognostic significance.

2. For the multivariate analysis of the entire cohort, it would be important to show the significance of mitotic rate (using the best individual cutpoint or the index mentioned in point 1 above) along with the other factors analyzed, so that its impact can be understood in the context of the other factors available.

3. As indicated above, there are significant limitations to the cohorts available that may preclude additional significant observations regarding the prognostic significance of mitotic rate in this analysis. One limitation is the preponderance of stage I cases (76% of the cohort), which in part accounts for 50% of the cases having a mitotic index of 0. Therefore, the proportion of cases with elevated mitotic rate is low. In addition, this results in smaller sample sizes for T3 and T4 cases, precluding a meaningful assessment of the prognostic impact in these subsets. These issues should be explicitly discussed.

4. What is the median follow up of the two cohorts? This should be explicitly stated, as it represents another major limitation of the study, especially in view of the predominance of stage I lesions, which can present with late relapses and deaths (i.e., beyond five years).

5. Table 1- 46% of patients were male. This is in contrast to most studies, which show a male predominance. Do the authors have any explanation for this finding?

6. PLOS authors have the option to publish the peer review history of their article (what does this mean?). If published, this will include your full peer review and any attached files.

Reviewer #1: No

Reviewer #2: No

---

## [Author Response · Author response to Decision Letter 0]

7 Dec 2023

Authors: Done.

2. Our staff editors have determined that your manuscript is likely within the scope of our Personalised Medicine Call for Papers. This call for papers aims to showcase the breadth of research within personalized medicine and its impact on innovating the healthcare horizon. Additional information can be found on our announcement page: Personalized Medicine Call for Paper- PLOS Collection

If you would like your manuscript to be considered for this collection, please let us know in your cover letter and we will ensure that your paper is treated as if you were responding to this call. Please note that being considered for the Collection does not require additional peer review beyond the journal’s standard process and will not delay the publication of your manuscript if it is accepted by PLOS ONE. If you would prefer to remove your manuscript from collection consideration, please specify this in the cover letter.

Authors: We would be honoured to have our article considered in the collection Personalized Medicine, please see our latest cover letter.

Authors: Done.

Authors: We made available the minimal anonymized data set necessary to replicate our findings at the link https://doi.org/10.6084/m9.figshare.24637938.v1 .

Additional Editor Comments :

The introduction has to be re-¬‐structured, and the rationale for the study should be set clear. The manuscript should also be checked for types and grammars errors.

Authors: Thank you for the suggestions. We have carefully reviewed the entire manuscript and had the text corrected by our language and editing service.

Comments to the Author

1. Is the manuscript technically sound, and do the data support the conclusions?

Reviewer #1: Yes

Reviewer #2: Yes

Authors: Thank you.

2. Has the statistical analysis been performed appropriately and rigorously?

Reviewer #1: Yes

Reviewer #2: Yes

Authors: Thank you.

3. Have the authors made all data underlying the findings in their manuscript fully available?

Reviewer #1: Yes

Reviewer #2: Yes

Authors: Thank you.

4. Is the manuscript presented in an intelligible fashion and written in standard English?

Reviewer #1: Yes

Reviewer #2: Yes

Authors: Thank you.

5. Review Comments to the Author

Reviewer #1: This is an interesting study looking at the prognostic role of Mitoses in melanoma patients. This topic has been discussed in the past years , since it gave indication to SLN biopsy in the last AJCC classification (upstaging the T1a to T1b). The study is well conducted and the results and discussion sound interesting. There authors made correction with SLNB status but not with the involvement of SLN (micro macro metastasis). Please have a look at Quaglino et al. Surg Oncol. 2011 Dec;20(4):259-64. Is there a particular reason not to look at this association? Please discuss.

Authors: Thank you for the positive feedback. We agree on the importance that SLN involvement could potentially have at the prognostic level. Unfortunately, micro/macro metastasis classification is not available in our registries. We have added this aspect in the study limitation paragraph properly citing Quaglino et al.

Reviewer #2: In this manuscript the authors examine the prognostic significance of mitotic rate in two cohorts of melanoma patients from the Veneto Cancer Registry. They show a correlation between increasing mitotic rate and various histopathologic factors in melanoma. They also show worse survival with increasing mitotic rate by Kaplan-Meier analysis. They investigate various cutoffs for mitotic rate in the entire cohorts, and in individual T categories. They conclude that mitotic rate does have prognostic significance and may be used in individualized prognostic assessments of melanoma patients. Overall, the manuscript is useful in its reinforcement of the importance of mitotic rate. Attention to the following points is warranted to strengthen their conclusions, and also to address limitations in the available cohorts that precluded additional observations regarding the prognostic significance of mitotic rate.

1. A critical issue regarding assessment of mitotic rate, given the non-linear shape of its impact on survival, is the identification of optimal cut-points, both in the entire cohort of patients, as well as in specific T categories, as has been previously demonstrated (ref 14). To begin with, this should be discussed in greater detail in the Discussion section. While the authors examine this issue in some analyses, additional analyses are suggested. For example, in their K-M analysis of the entire cohort in Fig. 1, the authors examine different cutpoints for mitotic rate (0, 1, 2, 3-5; and >6). This analysis suggests the following “optimal” grouping or index for mitotic rate with differing survival: 0-2; 3-5; and > 6. It would be both interesting and important to analyze this further, both in univariate and multivariate analyses for the entire cohort. Also, given that the analysis of individual cutpoints only showed significant results in T2 patients, it would be interesting to examine this index in each T category to determine whether it would show a significant prognostic significance.

Authors: Thanks for the interesting insight into the nonlinear form of MR impact on survival and challenges in optimal cut-points identification. We have added a paragraph regarding this in the discussion. In addition, we did further regression analysis to estimate the effects of mitotic count on survival using the “optimal” grouping 0-2; 3-5; and ≥6 (Table 3). The results, both using group encoding, both using single cut-offs for MR, are now reported for the entire sample and for each value of T, from T1 to T4 (Tables 3 and 4).

2. For the multivariate analysis of the entire cohort, it would be important to show the significance of mitotic rate (using the best individual cutpoint or the index mentioned in point 1 above) along with the other factors analyzed, so that its impact can be understood in the context of the other factors available.

Authors: Thank you for this important advice. Age and TNM stage were found to remain strong predictors of survival, but also mitotic rate, together with site, histology, and sex turned out to be statistically associated with prognosis. In Table 3 we have now added the HR estimates and p values for the adjustment variables. 

3. As indicated above, there are significant limitations to the cohorts available that may preclude additional significant observations regarding the prognostic significance of mitotic rate in this analysis. One limitation is the preponderance of stage I cases (76% of the cohort), which in part accounts for 50% of the cases having a mitotic index of 0. Therefore, the proportion of cases with elevated mitotic rate is low. In addition, this results in smaller sample sizes for T3 and T4 cases, precluding a meaningful assessment of the prognostic impact in these subsets. These issues should be explicitly discussed.

Authors: Thank you for pointing out this issue. We added the following sentence in the study limitation paragraph to address this aspect:

[…] being a population study, a further limitation of the present work is the underrepresentation of subjects with T3 or T4 melanoma at diagnosis. Low sample size may have hindered statistical significance and power in these subpopulations. In our region (Veneto, Italy), prevention campaigns over the past 30 years have increased population awareness about pigmented lesions and, together with early diagnosis, have indeed significantly reduced the incidence of advanced-stage melanomas. On the other hand, we believe that the analysis of a big group of T1 and T2 patients, such as our, allowed interesting observations on MR value, particularly for T2 stage […] could lead to the proposal of a MR based personalized planning […] for T2N0 patients (IB and IIA stage), including the highly debated adjuvant therapy [16].

4. What is the median follow up of the two cohorts? This should be explicitly stated, as it represents another major limitation of the study, especially in view of the predominance of stage I lesions, which can present with late relapses and deaths (i.e., beyond five years).

Authors: Thank you for the suggestion. In Table 1, we included the follow-up years statistics, and we added the following sentence in the results section: 

The overall median follow-up time was 3.82 years (min-max 0.01 - 6.00). Stratifyng by T stage it was 3.95, 3.75, 3.63, and 3.11 years respectively for T1, T2, T3, and T4 subjects. 

5. Table 1- 46% of patients were male. This is in contrast to most studies, which show a male predominance. Do the authors have any explanation for this finding?

Authors: The ASCO Cancer Net Editorial Board (03/2023) document on melanoma states that “The average age at diagnosis is 65. [...] Before age 50, more women are diagnosed with melanoma than men”. Concordantly, in a recent population-based study on Victorian Cancer Registry data (Australia), the authors report that before 50 years old, melanoma diagnosis is more prevalent in females (51.8%) over males (48.2%) (https://www.cancervic.org.au/research/vcr/cancer-fact-sheets/melanoma.html#how-common-is-melanoma). Similar results are also shown in the latest SEER update of delay-adjusted CMM incidence rate trends (https://seer.cancer.gov/statistics-network/explorer/application.html?site=53&data_type=1&graph_type=2&compareBy=sex&chk_sex_3=3&chk_sex_2=2&rate_type=2&race=1&age_range=9&stage=101&advopt_precision=1&advopt_show_ci=on&hdn_view=0&advopt_show_apc=on&advopt_display=2#resultsRegion0). The median age of our sample is 59 years (much less than 65, as above reported) and this might be an explanation for this finding.

---

## [Decision Letter · Decision Letter 1]

12 Mar 2024

PONE-D-22-26758R1Could the mitotic count improve personalized prognosis in melanoma patients?PLOS ONE

Dear Dr. Cozzolino,

Thank you for submitting your manuscript to PLOS ONE. After careful consideration, we feel that it has merit but does not fully meet PLOS ONE’s publication criteria as it currently stands. Therefore, we invite you to submit a revised version of the manuscript that addresses the points raised during the review process.

Please submit your revised manuscript within Apr 26 2024 11:59PM. If you will need more time than this to complete your revisions, please reply to this message or contact the journal office at plosone@plos.org. Please include the following items when submitting your revised manuscript:A rebuttal letter that responds to each point raised by the academic editor and reviewer(s). You should upload this letter as a separate file labeled 'Response to Reviewers'.A marked-up copy of your manuscript that highlights changes made to the original version. You should upload this as a separate file labeled 'Revised Manuscript with Track Changes'.An unmarked version of your revised paper without tracked changes. You should upload this as a separate file labeled 'Manuscript'.If applicable, we recommend that you deposit your laboratory protocols in protocols.io to enhance the reproducibility of your results. Protocols.io assigns your protocol its own identifier (DOI) so that it can be cited independently in the future. For instructions see: https://journals.plos.org/plosone/s/submission-guidelines#loc-laboratory-protocols. Additionally, PLOS ONE offers an option for publishing peer-reviewed Lab Protocol articles, which describe protocols hosted on protocols.io. Read more information on sharing protocols at https://plos.org/protocols?utm_medium=editorial-email&utm_source=authorletters&utm_campaign=protocols.

We look forward to receiving your revised manuscript.

Kind regards,

Dr H. Boukerche, PhD

Academic Editor

PLOS ONE

Journal Requirements:

Reviewers' comments:

Reviewer's Responses to Questions

**Comments to the Author**

1. If the authors have adequately addressed your comments raised in a previous round of review and you feel that this manuscript is now acceptable for publication, you may indicate that here to bypass the “Comments to the Author” section, enter your conflict of interest statement in the “Confidential to Editor” section, and submit your "Accept" recommendation.

Reviewer #3: All comments have been addressed

2. Is the manuscript technically sound, and do the data support the conclusions?

Reviewer #3: Yes

3. Has the statistical analysis been performed appropriately and rigorously? 

Reviewer #3: Yes

4. Have the authors made all data underlying the findings in their manuscript fully available?

Reviewer #3: Yes

5. Is the manuscript presented in an intelligible fashion and written in standard English?

Reviewer #3: Yes

6. Review Comments to the Author

Reviewer #3: The assessed manuscript was a 'Revised Manuscript with Track Changes' in the document PONE-D-22-26758_R1_reviewer. In general, the objective is clear, and the methodology is appropriate.

- In the abstract, the authors should add the time of overall survival evaluated (3-year overall survival) and use the term 'histological subtype' instead of 'type of morphology'.

- In Statistical Methods, make it clear that the considered period for short-term overall survival was 3 years.

- In Tables 1 and 2, it is important to include in the caption what 'Malignant (NOS)' means, once it is not a histological subtype of melanoma.

- In the title of Table 2, there is no need for the phrase “p-values < 0.05 are reported in bold”

- In Table 2, the percentage is not consistently represented (Some values have two numbers after the decimal point, others do not).

- Fig.1 is mentioned in the text, the title appears, but I could not find either the figure or its caption.

- Tables 3 and 4: highlight significant p-values in the adjusted model using bold, not red color.

- In the discussion, it is more appropriate to write "increased mitoses per mm2 are associated with a higher risk of mortality, particularly in T2 patients," instead of "mitoses per mm2 are associated with an increased risk of mortality, particularly in T2 patients."

- In the discussion, it is more appropriate to write "In particular, we observe that patients with a high mitotic rate are more likely to be older, male, have a melanoma lesion with a vertical growth pattern, nodular histological subtype, ulceration, greater tumor thickness, and an advanced stage," instead of "In particular, we observe that patients with a high mitotic rate are more likely to be older, male, have a melanoma lesion with a vertical growth pattern, ulceration, greater tumor thickness, and an advanced stage."

- In the discussion, "... the presence of pre-existing nevi were found to be typical of lesions with sparse mitotic activity," however, there is no information about the absence or presence of pre-existing nevi in the results. Please, insert the data or remove this sentence.

- In the discussion, reference 14 appears in the wrong place (page 72 of the document PONE-D-22-26758_R1_reviewer).

- In the last paragraph of the discussion, it should be T2N0M0, not T2N0.

7. PLOS authors have the option to publish the peer review history of their article (what does this mean?). If published, this will include your full peer review and any attached files.

Reviewer #3: No

---

## [Author Response · Author response to Decision Letter 1]

13 Mar 2024

Authors’ response to Review Comments to the Author

Reviewer #3: The assessed manuscript was a 'Revised Manuscript with Track Changes' in the document PONE-D-22-26758_R1_reviewer. In general, the objective is clear, and the methodology is appropriate.

• In the abstract, the authors should add the time of overall survival evaluated (3-year overall survival) and use the term 'histological subtype' instead of 'type of morphology'.

Authors: Ok, done. Thank you.

• In Statistical Methods, make it clear that the considered period for short-term overall survival was 3 years.

Authors: Ok, done. Thank you.

• In Tables 1 and 2, it is important to include in the caption what 'Malignant (NOS)' means, once it is not a histological subtype of melanoma.

 Authors: Ok, abbreviations meaning are now specified in all the tables. Thank you.

• In the title of Table 2, there is no need for the phrase “p-values < 0.05 are reported in bold” 

Authors: Ok, deleted. Thank you.

• In Table 2, the percentage is not consistently represented (Some values have two numbers after the decimal point, others do not).

Authors: Zeros were originally omitted, resulting in inconsistent representation (number of digits). Now all estimates are reported with 2 digits after decimal point. Thank you.

• Fig.1 is mentioned in the text, the title appears, but I could not find either the figure or its caption.

Authors: Images were not included in the main manuscript as per PLOS ONE guidelines. Only figure captions were originally shown. Fig 1 and caption are now clearly reported in the Results section.

Tables 3 and 4: highlight significant p-values in the adjusted model using bold, not red color.

Authors: Ok, done. Thank you.

• In the discussion, it is more appropriate to write "increased mitoses per mm2 are associated with a higher risk of mortality, particularly in T2 patients," instead of "mitoses per mm2 are associated with an increased risk of mortality, particularly in T2 patients."

Authors: Amended as suggested, thank you.

• In the discussion, it is more appropriate to write "In particular, we observe that patients with a high mitotic rate are more likely to be older, male, have a melanoma lesion with a vertical growth pattern, nodular histological subtype, ulceration, greater tumor thickness, and an advanced stage," instead of "In particular, we observe that patients with a high mitotic rate are more likely to be older, male, have a melanoma lesion with a vertical growth pattern, ulceration, greater tumor thickness, and an advanced stage."

Authors: Amended as suggested, thank you.

• In the discussion, "... the presence of pre-existing nevi were found to be typical of lesions with sparse mitotic activity," however, there is no information about the absence or presence of pre-existing nevi in the results. Please, insert the data or remove this sentence.

Authors: We rephrased the sentence. Thank you for the correction. 

• In the discussion, reference 14 appears in the wrong place (page 72 of the document PONE-D-22-26758_R1_reviewer).

Authors: We thoroughly checked references citation numbers. Thank you for pointing out this error.

• In the last paragraph of the discussion, it should be T2N0M0, not T2N0.

Authors: Corrected. Thank you.

---

## [Editor Report · Decision Letter 2]

2 Apr 2024

Could the mitotic count improve personalized prognosis in melanoma patients?

PONE-D-22-26758R2

Dear Dr. Claudia Cozzolino,

We’re pleased to inform you that your manuscript has been judged scientifically suitable for publication and will be formally accepted for publication once it meets all outstanding technical requirements.

Kind regards,

Dr H.  Boukerche, PhD

Academic Editor

PLOS ONE

---

## [Editor Report · Acceptance letter]

4 Apr 2024

PONE-D-22-26758R2 

PLOS ONE

Dear Dr. Cozzolino, 

I'm pleased to inform you that your manuscript has been deemed suitable for publication in PLOS ONE. Congratulations! Your manuscript is now being handed over to our production team.

Kind regards, 

on behalf of

Dr Habib Boukerche 

Academic Editor

PLOS ONE